# The cancer angiogenesis co-culture assay: *In vitro* quantification of the angiogenic potential of tumoroids

**Sarah Line Bring Truelsen**[1], **Nabi Mousavi**[2], **Haoche Wei**[3], **Lucy Harvey**[3], **Rikke Stausholm**[1], **Erik Spillum**[4], **Grith Hagel**[1], **Klaus Qvortrup**[5], **Ole Thastrup**[1], **Henrik Harling**[1,6], **Harry Mellor**[3], **Jacob Thastrup**[1]*

1 2cureX, Symbion, Copenhagen, Denmark, 2 Department of Pathology, Rigshospitalet, University of Copenhagen, Copenhagen, Denmark, 3 School of Biochemistry, University of Bristol, Bristol, United Kingdom, 4 BioSense Solutions, Farum, Denmark, 5 Department of Biomedical Sciences, University of Copenhagen, Copenhagen, Denmark, 6 Department of Digestive Diseases, Bispebjerg Hospital, University of Copenhagen, Copenhagen, Denmark

* jt@2curex.com

**Data Availability Statement:** All relevant data are within the paper and its Supporting Information files.

## Abstract

The treatment response to anti-angiogenic agents varies among cancer patients and predictive biomarkers are needed to identify patients with resistant cancer or guide the choice of anti-angiogenic treatment. We present "the Cancer Angiogenesis Co-Culture (CACC) assay", an *in vitro* Functional Precision Medicine assay which enables the study of tumouroid induced angiogenesis. This assay can quantify the ability of a patient-derived tumouroid to induce vascularization by measuring the induction of tube formation in a co-culture of vascular cells and tumoroids established from the primary colorectal tumour or a metastasis. Furthermore, the assay can quantify the sensitivity of patient-derived tumoroids to anti-angiogenic therapies. We observed that tube formation increased in a dose-dependent manner upon treatment with the pro-angiogenic factor vascular endothelial growth factor A (VEGF-A). When investigating the angiogenic potential of tumoroids from 12 patients we found that 9 tumoroid cultures induced a significant increase in tube formation compared to controls without tumoroids. In these 9 angiogenic tumoroid cultures the tube formation could be abolished by treatment with one or more of the investigated anti-angiogenic agents. The 3 non-angiogenic tumoroid cultures secreted VEGF-A but we observed no correlation between the amount of tube formation and tumoroid-secreted VEGF-A. Our data suggests that the CACC assay recapitulates the complexity of tumour angiogenesis, and when clinically verified, could prove a valuable tool to quantify sensitivity towards different anti-angiogenic agents.

## Introduction

Impaired control of angiogenesis is involved in numerous pathological conditions and is a hallmark of cancer progression and metastasis [1–3]. Solid tumours are constantly influenced by the distance to nearest capillary and growth beyond a diameter of 2 mm requires

**Funding:** SLBT and NM was supported by the Innovation Fond Denmark http://innovationsfonden.dk/en), grant no. 5184-00101B. SLBT received funding from Agnes & Poul Friis Fund (grant no. 81008-001), C.C. Klestrup & hustrus Henriette Klestrups Mindelegat (grant no. 0660-001), Carl & Ellen Hertz legat til dansk læge- og naturvidenskab (grant no. 7179-2), The Drost Fundation (grant no. 12120-1), and Familien Hede Nielsens Fund (grant no 2017-0423). The funders provided support in the form of salaries for authors SLBT and NM but did not have any additional role in the study design, data collection and analysis, decision to publish, or preparation of the manuscript. The specific roles of these authors are articulated in the 'author contributions' section. SLBT, GH, OT, RS and JT are fulltime employees at 2cureX, Symbion, Copenhagen, Denmark (https://www.2curex.com/). 2cureX provided support in the form of salaries for authors GH, OT, RS and JT. ES is a fulltime employee at BioSense Solutions, Farum, Denmark (https://biosensesolutions.dk/). BioSense Solutions provided support in the form of salaries for ES. HW, LH, KQ, HH and HM received no specific funding for this work.

**Competing interests:** I have read the journal's policy and the authors of this manuscript have the following competing interests: SLBT, GH, OT, RS and JT are fulltime employees at 2cureX, Symbion, Copenhagen, Denmark. GH and OT are founders and stockholders for 2cureX. ES is a fulltime employee at BioSense Solutions, Farum, Denmark. ES is a founder and stockholders for BioSense Solutions. This does not alter our adherence to PLOS ONE policies on sharing data and materials. NM, HW, LH, KQ, HH and HM declare no competing interests.

vascularization of the tumour to facilitate continuous diffusion of oxygen and nutrients [4]. This neovascularization occurs by several mechanisms including sprouting angiogenesis and vascular co-option, where cancer cells grow along existing vessels in the peri-tumoural tissue and integrate them in the expanding tumour [5]. Histological analyses of premalignant, non-invasive lesions, showed that angiogenesis is an early event during the development of invasive cancers [6]. Formation of new vessels includes endothelial cell proliferation and migration followed by assembly into a new vascular structure, lumen formation and, finally, maturation of the newly formed endothelial tube [5]. Neovascularization is initiated by changes in the balance between pro- and anti-angiogenic molecules in the microenvironment of the tumour [6]. The most potent pro-angiogenic factor under both physiological and pathological conditions is vascular endothelial growth factor (VEGF)-A [7].

Colorectal cancer is the fourth leading cause of cancer-related deaths worldwide [8]. At time of diagnosis, approximately 20% of colorectal cancer patients have synchronous metastases. In total, close to 50% develop metastases at some point during the clinical course of their disease [9]. Treatment of metastatic colorectal cancer (mCRC) often includes chemotherapeutics and a molecular targeted agent, such as an anti-angiogenic agent [10]. Anti-angiogenic therapy can normalize the tumour vasculature by interfering with angiogenic ligands, their receptors or their downstream signalling [11]. Normalization is obtained when the equilibrium between pro- and anti-angiogenic factors is restored, typically within 1–2 days after treatment initiation. Effective anti-angiogenic treatment enhances the vascular circulation and decreases the interstitial pressure of the tumour which improves the delivery of both nutrients and chemotherapeutics [12].

The first anti-angiogenic agent to be approved by the FDA in malignancies was the ligand trapping humanized monoclonal antibody bevacizumab [13, 14]. Bevacizumab binds to and neutralizes all human VEGF-A isoforms and bioactive proteolytic fragments [15, 16]. A pooled analysis from seven randomized controlled trials in mCRC showed that addition of bevacizumab to a chemotherapeutic backbone increased median overall survival (OS) from 16.1 to 18.7 months (p = 0.0003) and progression free survival from 6.4 to 8.8 months (p<0.0001) [17]. Several anti-angiogenic agents have subsequently been approved for the treatment of various malignancies, including the first-generation tyrosine kinase inhibitors (TKIs) sorafenib and sunitinib and the second generation TKI regorafenib. Sorafenib was FDA approved for the treatment of advanced renal cell carcinoma in 2005 and sunitinib received FDA approval for the treatment of gastrointestinal stromal tumours and advanced renal cell carcinoma in 2006 [18, 19]. Regorafenib was approved as monotherapy in third-line treatment of mCRC after studies showed a significant increase in OS from 5.9 months in the placebo group to 6.4 months in the regorafenib group (p = 0.0052) [20, 21]. The TKIs are pan-kinase inhibitors, but all of them target VEGFR-2, platelet-derived growth factor receptor β (PDGFRβ) and in the case of regorafenib and sorafenib fibroblast growth factor receptor (FGFR) [22–24].

Numerous studies have focused on Precision Medicine in order to optimize treatment, improve outcome and avoid undesired adverse effects of ineffective treatments for each patient [25–31]. Suggested predictive biomarkers for anti-angiogenic treatment include intratumoural and plasma VEGF-A levels, tumour imaging, histopathological growth patterns, pro-inflammatory cytokines, soluble VEGF receptors, gene signatures, and polymorphisms in the VEGF-A pathway genes. Most of these biomarkers were suggested on the basis of results from small patient series or retrospective data, and they all lack clinical validation [11, 32, 33].

In the effort of personalizing anti-angiogenic treatment an *in vitro* sensitivity test could be an alternative to the aforementioned predictive biomarkers. Accordingly, tumour cells from the individual patient could be introduced into an *in vitro* angiogenesis assay and evaluated for their angiogenic potential and sensitivity to anti-angiogenic agents. Such an assay should be

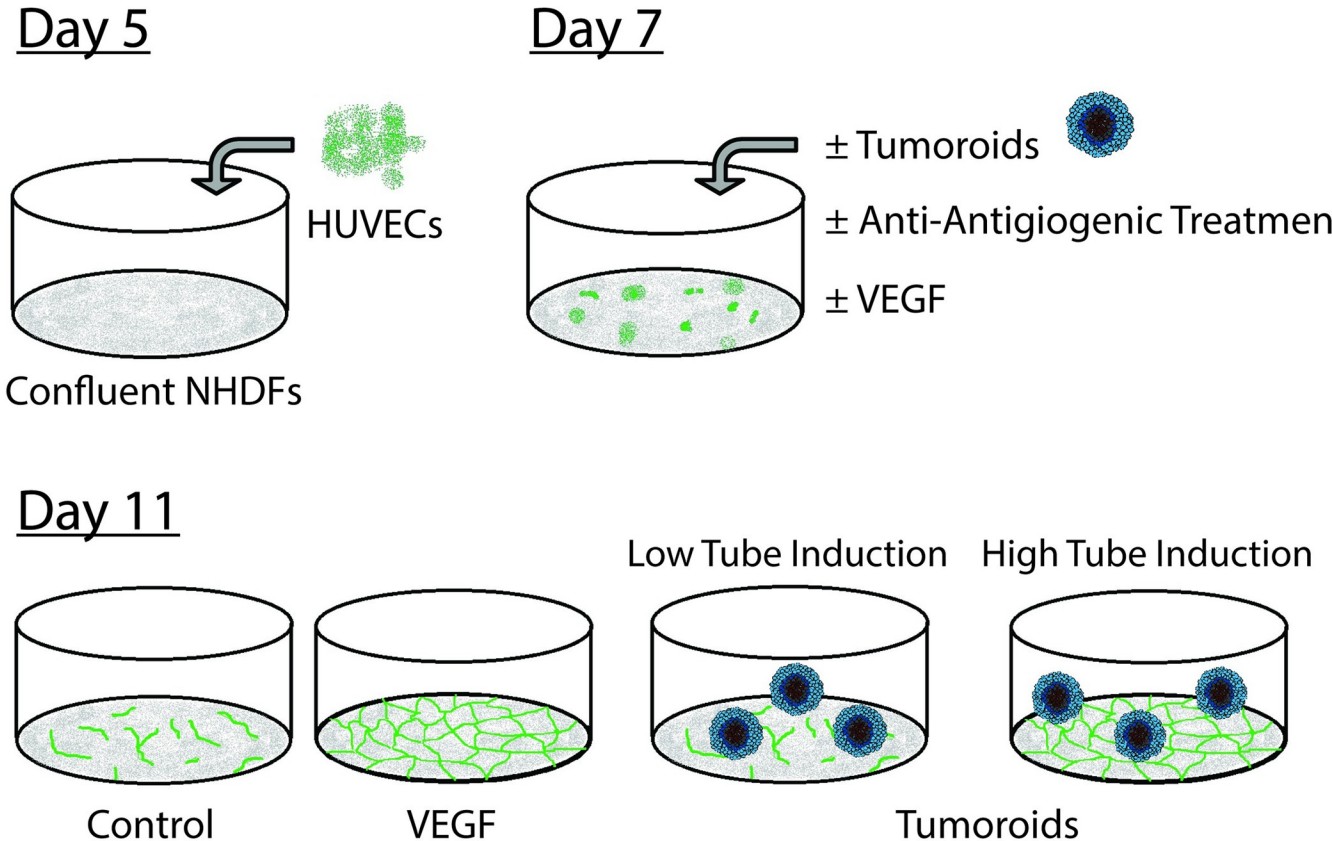

**Fig 1. Overview of the cancer angiogenesis co-culture (CACC) assay.** (DAY 5) HUVECs were seeded on a confluent layer of NHDFs. (DAY 7) Following two days of HUVEC-NHDF co-culture tumoroids/human recombinant VEGF-A$_{165}$ (VEGF)/anti-angiogenic treatments were introduced to the assay in a 1:4 dilution of GFR Matrigel. (DAY 11) Samples were fixed and stained with the endothelial marker PECAM-1 to visualize tubes.

able to evaluate the combined contribution of both pro- and anti-angiogenic factors secreted by the tumour [34, 35].

In this study we present our novel Cancer Angiogenesis Co-Culture (CACC) Assay (Fig 1). The assay is a modification of the organotypic co-culture assay of angiogenesis developed by Bishop *et al*. in which human umbilical vein endothelial cells (HUVECs) are co-cultured with juvenile foreskin fibroblasts for 14 days, during which time the endothelial cells (ECs) formes luminal tubules resembling capillaries *in vivo* [36]. The assay recapitulates the key aspects of angiogenesis such as extension of sprouts, branching of existing structures, migration by cord tip cells and tubulogenesis [36–38]. The assay modifications presented in this study have allowed for co-culture of fibroblasts, vascular cells, and tumoroids established from primary colorectal tumours or metastases and thereby enabled the study of tumour induced angiogenesis.

## Materials and methods

### Culture of cell lines

Primary HUVECs were obtained from Lonza (Basel, Switzerland) and cultured in endothelial cell growth medium-2 (EGM-2). EGM-2 medium consisted of endothelial basal medium (EBM) supplemented with EGM-2 SigleQuot (Lonza) or SupplementPack (PromoCell,

Heidelberg, Germany) containing 2% fetal bovine serum (FBS; Sigma-Aldrich, St. Louis, MO, USA), endothelial growth factor (EGF), insulin-like growth factor (IGF), human fibroblast growth factor 2 (bFGF), heparin, ascorbic acid, hydrocortisone, gentamicin, and amphotericin-B. EBM was either provided from Lonza, PromoCell or mixed by adding 1% fatty acid free (FAF) bovine albumin serum (BSA; Sigma-Aldrich) to Dulbecco's modified eagles medium (DMEM)/F12 (Thermo Fisher Scientific, Waltham, MA, USA). Prior to HUVEC seeding, the culture flasks were coated with 10 µg/ml fibronectin (Sigma-Aldrich) in phosphate-buffered saline (PBS; Sigma-Aldrich) for 1h at room temperature (RT). Primary adult normal human dermal fibroblasts (NHDFs; Batch A: Lonza, Batch B: PromoCell) and the colon cancer cell line SW480 were cultured in DMEM (Sigma-Aldrich) with 10% FBS supplemented with 100 U/ml penicillin and 100 µg/ml streptomycin (P/S; Sigma-Aldrich). HUVEC and NHDF were passaged every 2–3 days with split ratios 1:4–6. For experiments HUVEC cells were used in passage 6–8 and NHDF in passage 8–10. All cell lines were cultured at 37˚C in a humidified atmosphere with 5% $CO_2$.

## Preparation of a VEGF-expressing SW480 cancer cell line

A VEGF-expressing SW480 cell line was generated by lentiviral transduction. Human VEGF-A$_{165}$ (VEGF) was cloned into the lentiviral expression vector pLVX-Puro (Takara Bio, Gothenburg, Sweden). The expression vector was co-transfected with pMD2.G (AddGene, Watertown, MA, USA) and psPAX2 (AddGene) into Lenti-X™ 293T packaging cells (Takara Bio) using PEI (Sigma Aldrich) at a final concentration of 1uM. The produced virus particles were harvested 48 and 72hrs post transfection and filtered through a 0.4um filter. The SW480 cells were transduced with the produced viruses and a stable population of VEGF expressing cells was isolated by selection with puromycin (2ug/ml for 48h, Thermo Fisher Scientific).

## Secretion of angiogenic factors from cancer cell lines and tumoroids

The concentration of VEGF-A, bFGF and PDGF-BB was directly measured in culture supernatants using enzyme-linked immunosorbent assay (ELISA) kits following the manufacturer's recommendations. For cell line measurements R&D Systems (Minneapolis, MN, USA) and for tumoroids Thermo Fisher Scientific (VEGF and bFGF) and R&D Systems (PDGF-BB). Tumoroids were cultured in the CACC assay and supernatants collected on day 11. VEGF-expressing SW480 cells (SW480-V) and the parental line were cultured to confluence and allowed to condition the cell media for 48h. VEGF secretion was also examined qualitatively by immunofluorescence microscopy. Briefly, cells were fixed with 4% paraformaldehyde for 15 min and permeabilized with 0.5% Triton X-100 for 5 min. Cells were incubated with VEGF antibody (R&D systems) for 1h, followed by secondary Alexa-488 conjugated goat anti-mouse antibody (Invitrogen Carlsbad, CA, USA) for 45 min. Accordingly, the cells were incubated with DAPI for 5 min and mounted in Mowiol (Sigma-Aldrich) containing 2.5% 1,4-diazabicyclo[2.2.2]octane (DABCO) as an anti-bleaching agent. Images were obtained using a Leica SP5-AOBS confocal laser scanning microscope attached to a Leica DM I6000 inverted epi-fluorescence microscope.

## Formation of cell line spheroids

SW480 and SW480-V cells (1.5 x 10$^5$ cells) were suspended in 3 ml of complete DMEM medium and mixed with 1.5 ml of a sterile suspension of methylcellulose (2.4% in DMEM). The suspended cells were dispensed in drops of 30 µl onto the lids of 10 cm culture dishes which were inverted in a humidified incubator for seven days. Subsequently, spheroids were harvested by suspension in PBS and gentle centrifugation at 160 x g for 90s. Next, the

spheroids were resuspended in an ice-cold mixture of 30% Geltrex (Thermo Fisher Scientific) in complete EGM-2 medium (Lonza), for addition to the CACC assay. To visualize vessel formation the cells were fixed and stained with human anti-PECAM-1 antibody (R&D systems) in 1% BSA (Sigma-Aldrich) in PBS (1h, 37°C), followed by Alkaline phosphatase-conjugated secondary antibody (Novus Biologicals, Littleton, CO, USA) diluted in 1% BSA in PBS (1h, 37°C) and treated with the substrate SigmaFAST BCIP/NBT (5-bromo-4-chloro-3-indolyl phosphate/nitro blue tetrazolium; Sigma-Aldrich) for 15–30 min at 37°C. Tubes were imaged by widefield light microscopy.

## Patient samples

Fresh tumour tissue was obtained from either surgically resected colonic tumours and liver metastases before routine histological processing of the specimen or by ultrasound-guided needle biopsies. The study protocols were approved by the Regional Committee on Health Research Ethics—Capital Region of Denmark (protocol no. H-1-2011-125 and H-16031106) and informed consent was obtained from all patients. Samples from primary tumours were collected at Bispebjerg Hospital, Denmark and samples from liver metastases were collected at Rigshospitalet, Denmark. Tumour tissue was placed in cold PBS supplemented with 2.5 μg/ml amphotericin B (Sigma-Aldrich) and antibiotics (500 U/ml and 500 μg/ml P/S and 100 μg/ml gentamicin (Sigma-Aldrich)) and subsequently transported to the laboratory on ice. For each patient age, gender, and TNM stage were recorded.

## Establishment of patients-derived tumoroids

Patient-derived tumoroids were established as described by Jeppesen *et al*. [39] with minor modifications. In short, freshly resected tumour samples of 0.01–1 g were divided into 1 mm$^3$ pieces and digested for 20 min at 37°C with 1 mg/ml collagenase type II (Thermo Fisher Scientific) diluted in STEM medium. Subsequently, the tissue suspension was sequentially filtered through a 70 μm cell strainer (VWR, Soeborg, Denmark) and a 30 μm pre-separation filter (MACS, Miltenyi Biotec, Bergisch Gladbach, Germany). Tissue retained by the 70 μm filter was collected and redigested for 10 min at 37°C and passed through the filters again. This step was repeated until all tissue could pass through the 70 μm filter. Retained tumour fragments (30–70 μm) were seeded in STEM cell medium in petri dishes coated with 1.5% agarose (Sigma-Aldrich) and cultured at 37°C in a 5% $CO_2$ humidified incubator. After 3–5 days the tissue fractions which had densified into rounded tumoroids with smooth surfaces could be collected for further culture. Tumoroids established from colonic tumours were denoted C after the patient number and tumoroids from liver metastases were denoted L.

## Propagation and culture of patients-derived tumoroids

To increase the size and number of tumoroids per patient, tumoroids sized 30–70 μm were seeded in drops of 50 μl Matrigel matrix (VWR) diluted 1:2 with STEM cell medium in a 24-well dish. Following solidification (30 min, 37°C), in a 5% $CO_2$ humidified incubator, 1 ml STEM cell medium was added to each well. After an average of 2 weeks, with intermediate medium change, the tumoroids exceeding 200 μm in size could be collected for experiments with a 200 μm pluristrainer (PluriSelect, Leipzig, Germany). Tumoroids used in this study were thawed from the 2cureX tumoroid biobank and cultured as described above before experiments.

## Cancer angiogenesis co-culture assay

In the original co-culture assay of angiogenesis, established by Bishop *et al.* [36] and later modified by Mavria and colleagues [40], endothelial cells (HUVECs) are cultured in EGM-2 medium in co-culture with fibroblasts. Under these conditions the HUVECs form tube-like structures within the layers of fibroblasts and extra cellular matrix [36, 37, 40]. The Cancer Angiogenesis Co-Culture (CACC) Assay established for our studies follow the same procedure until day 7 after which the culture condition has been modified, since patient-derived tumoroids dissociate when cultured in EGM-2 medium. In short, NHDF were seeded in EGM-2 medium in 24-well plates at a density of $1.5–1.75 \times 10^4$ cells/well (batch and passage dependent density) and allowed to grow to confluence over five days with intermediate medium change. On day 5 the HUVEC cells were seeded onto the confluent layer of NHDF in EGM-2 medium at a density of $1.5 \times 10^4$ cells/well. After two days of HUVEC-NHDF co-culture (day 7) tumoroids were introduced to the assay (Fig 1). The tumoroids were isolated from the culture plates by gentle pipetting with cut-off P-1000 tips followed by filtration through a 400 μm and a 200 μm pluristrainer filter (PluriSelect). The 200–400 μm fraction of tumoroids were suspended in PBS and allowed to sediment without centrifugation. Subsequently, for each well 75 tumoroids were resuspended in 250 μl ice-cold matrix solution: Growth factor reduced (GFR) Matrigel (VWR) diluted 1:4 with reduced F12 medium (R-F12) consisting of DMEM/F12, 5% FBS, 0.45% BSA, 0.5% FAF BSA, 0.1 mM 2-mercaptoethanol, 100 U/ml and 100 μg/ml P/S, 50 μg/ml gentamicin, and 2.5 μg/ml amphotericin B. The tumoroid solution was carefully distributed on top of the HUVEC-NHDF co-culture and incubated at 37˚C for 30 min to ensure solidification of the matrix solution. Subsequently, 250 μl R-F12 medium ± treatment compounds were gently added on top of the matrix. Medium ± treatment was changed on day 9, and on day 11 the cells were fixed in 10% neutral buffered formalin containing 4% formaldehyde solution (Hounisen, Skanderborg, Denmark) for 2.5h at 4˚C with intermediate change of formalin after 30 min. To quantify vessel formation the cells were stained with human anti-PECAM-1 antibody (R&D systems) diluted in 1% BSA in PBS (1h, 37˚C) followed by Alexa Fluor 488 goat anti-mouse secondary antibody (Invitrogen, 1:500 in 1% BSA in PBS) for 1h at 37˚C and counterstained with Hoechst (Sigma, 5 μM in PBS) for 5 min at RT. Treatment compounds included VEGF-A$_{165}$, bevacizumab, regorafenib (Selleck Chemicals, Houston, TX, USA), sorafenib (Selleck Chemicals), and sunitinib (Selleck Chemicals). Each experiment included positive controls obtained by treatment with 10 ng/ml VEGF-A$_{165.}$ Microscope images (5x objective) of the wells were obtained with Cellomics Array Scanner V$^{TI}$ and a minimum of 8 representative images per sample were analysed with the proprietary BioSense Solutions and 2cureX tube algorithm.

## Detection of cell death in the CACC assay

To investigate the induction of cell death upon treatment with anti-angiogenic compounds the fixed culture was stained with human/mouse active caspase-3 antibody (R&D systems) together with human anti-PECAM-1 antibody (R&D systems) diluted with 1% BSA and 1% goat serum in PBS (1h, 37˚C), followed by the secondary antibodies Alexa Fluor 488 goat anti-mouse (1:500) and Alexa Flour 546 Goat anti-Rabbit (Invitrogen) for 1h at 37˚C. Subsequently, the samples were counterstained with Hoechst (5 min, RT, 5 μM in PBS). Positive controls were obtained by treatment with 1 μM staurosporine (Sigma-Aldrich) for 6h prior fixation. Microscope images (10x objective) were obtained with Cellomics Array Scanner V$^{TI}$ and a minimum of 10 representative images per sample were quantified with a proprietary 2cureX algorithm. The expression of active caspase-3 was investigated in areas covered with HUVEC cells (Active casp-3 masked Pecam-1) and in the entire well, including areas only covered by NHDFs (total casp-3 activation).

## Statistical analysis

Data are presented as mean ± standard deviation (SD) or standard error of the mean (SEM) and analysed by two-tailed unequal variance t-test using Excel (Microsoft, Seattle, CA, USA). P-values less than 0.05 were considered statistically significant.

# Results

## VEGF treatment increased tube formation in the CACC assay

To verify that the HUVECs retained the ability to form tubes and be stimulated by pro-angiogenic factors in the modified culture conditions, we treated with increasing concentrations of VEGF-A. In the untreated controls HUVECs grew in short tubes with only few branches and covered less than 10% of the investigated area (Fig 2A). VEGF-A treatment resulted in a dose-dependent induction of tube formation. Accordingly, addition of 10 ng/ml VEGF-A induced a two-fold increase in tube area compared to the untreated control (Fig 2B). Treatment with VEGF-A also increased branching.

## VEGF-induced tube formation was inhibited by anti-angiogenic compounds

Treatment with known anti-angiogenic inhibitors resulted in a significant inhibition of tube formation at concentrations from 10 ng/ml bevacizumab ($p < 0.05$), 25 nM regorafenib ($p < 0.05$), 10 nM sorafenib ($p < 0.001$), and 10 nM sunitinib ($p < 0.01$) when compared with VEGF-A treated samples (3.75 ng/ml VEGF-A$_{165}$ without anti-angiogenic compounds) (Fig 3A–3C). 3.75 ng/ml VEGF resembled the level observed in the tumoroid cultures with the highest amount of VEGF-secretion (patient 11-L and 12-L, Fig 5A). In addition, we found that concentrations from 50 nM sorafenib ($p < 0.005$) and 25 nM sunitinib ($p < 0.05$) resulted in a significant inhibition of tube formation when compared with vehicle treated controls (no VEGF-A).

## Tube formation concentrated around VEGF secreting spheroids

To investigate if angiogenic factors secreted from cancer cells grown in co-culture with HUVEC cells affect tube formation we transduced the colon cancer cell line SW480 with Human VEGF-A$_{165}$ (Fig 4). Microscopic imaging of VEGF stained cell monolayers revealed no detectable VEGF in the non-transduced cells (SW480). In contrast, the VEGF transduced SW480 cells (SW480-V) contained VEGF-positive vesicles and a diffuse pattern of VEGF staining beneath the cells was observed (Fig 4A). The SW480-V cells secreted almost 10 times more VEGF than the parental SW480 as determined by ELISA (Fig 4B). Finally, in the co-cultures we found that tube formation was directed towards the VEGF secreting spheroids (Fig 4C).

Sphere size affects the level of accessible oxygen throughout the sphere and thereby also the level of the *vegfa* transcription factor hypoxia-inducible factor α (HIF α) [41–43]. We investigated the correlation between tumoroid diameter and VEGF secretion for three tumoroid cultures. The results showed a 35-fold increase in average VEGF secretion between small (70-100um) and large (200-400um) tumoroids (S1 Fig).

## Patient-derived tumoroids induced tube formation

Ten different biobank cultures of large tumoroids (200–400 μm) were assessed for their ability to induce tube formation (Table 1). We found that 7 tumoroid cultures induced a significant

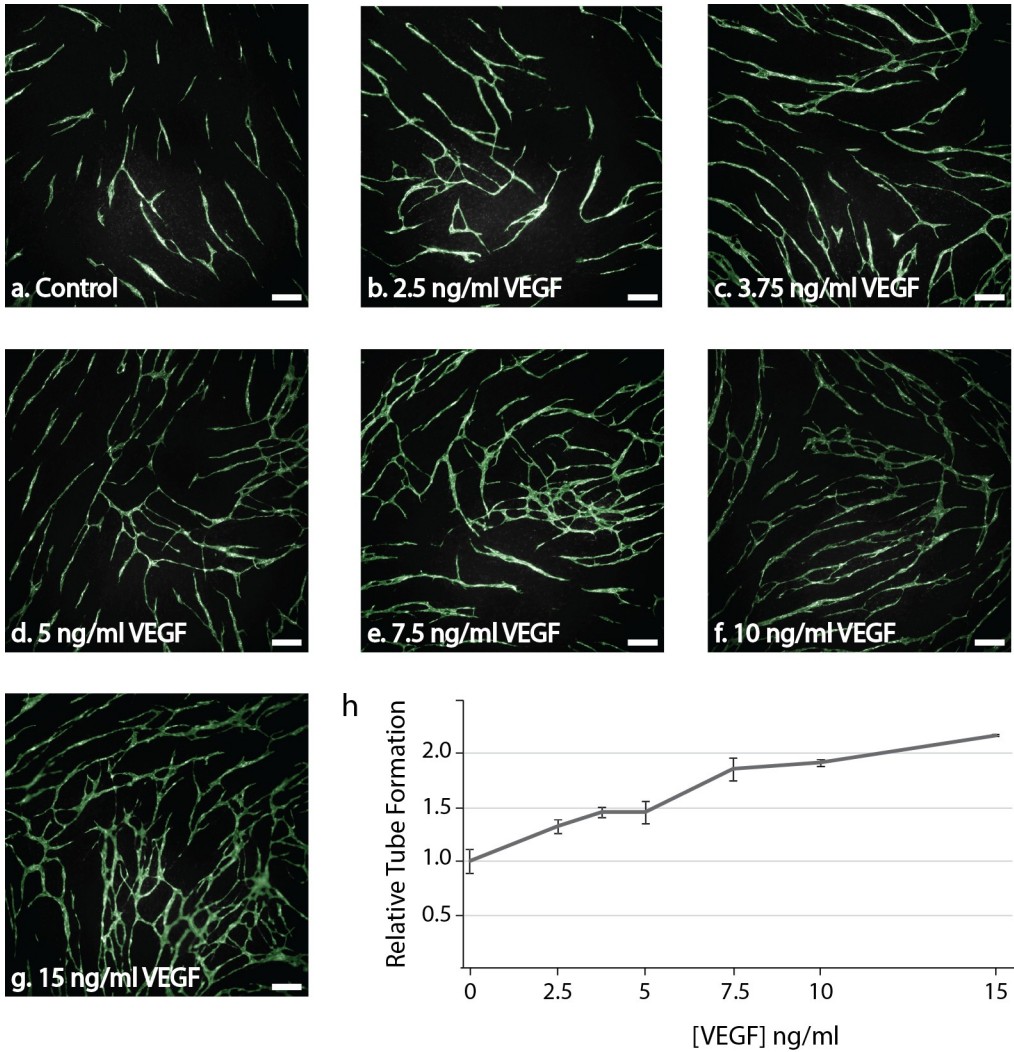

**Fig 2. Dose-response curve following treatment with the pro-angiogenic VEGF-A$_{165}$ (VEGF).** HUVECs were seeded on a confluent layer of NHDF on day 5 and cultured with increasing concentrations of VEGF (0–15 ng/ml) from day 7. At the end of the experiment (day 11), the co-cultures were fixed and stained for the endothelial marker PECAM-1 to reveal the tubes. (a-g) Representative fluorescent images (5x) obtained with the Array Scanner V$^{TI}$ with green overlay from the tube algorithm of PECAM-1 positive cells following treatment with (a) 0 ng/ml VEGF (Control), (b) 2.5 ng/ml VEGF, (c) 3.75 ng/ml VEGF, (d) 5 ng/m VEGF, (e) 7.5 ng/ml VEGF, (f) 10 ng/ml VEGF or (g) 15 ng/ml VEGF. Bar 200 µm. (h) Quantification of tube area in samples treated with increasing concentrations of VEGF. Tubes were quantified with the BioSense Solutions and 2cureX tube algorithm. Two independent experiments were performed with representative results displayed. Data are the average of triplicate samples ± SD.

(p<0.05) increase of tube formation (Fig 5A) and these were accordingly classified as angiogenic. Interestingly, the two cultures originating from the same patient (pt8) had similar increases in the level of tube formation. The highest induction of tubes occurred in patient 10-L showing a 1.43-fold increase compared to the control without tumoroids (p = 0.01). Tumoroids originating from this patient also induced a high amount of tube branching (Fig 5B) compared to the control. Tumoroids deriving from patient 5-C, 11-L and 12-L did not result in a statistically significant increased tube formation, and such cultures were therefore classified as non-angiogenic.

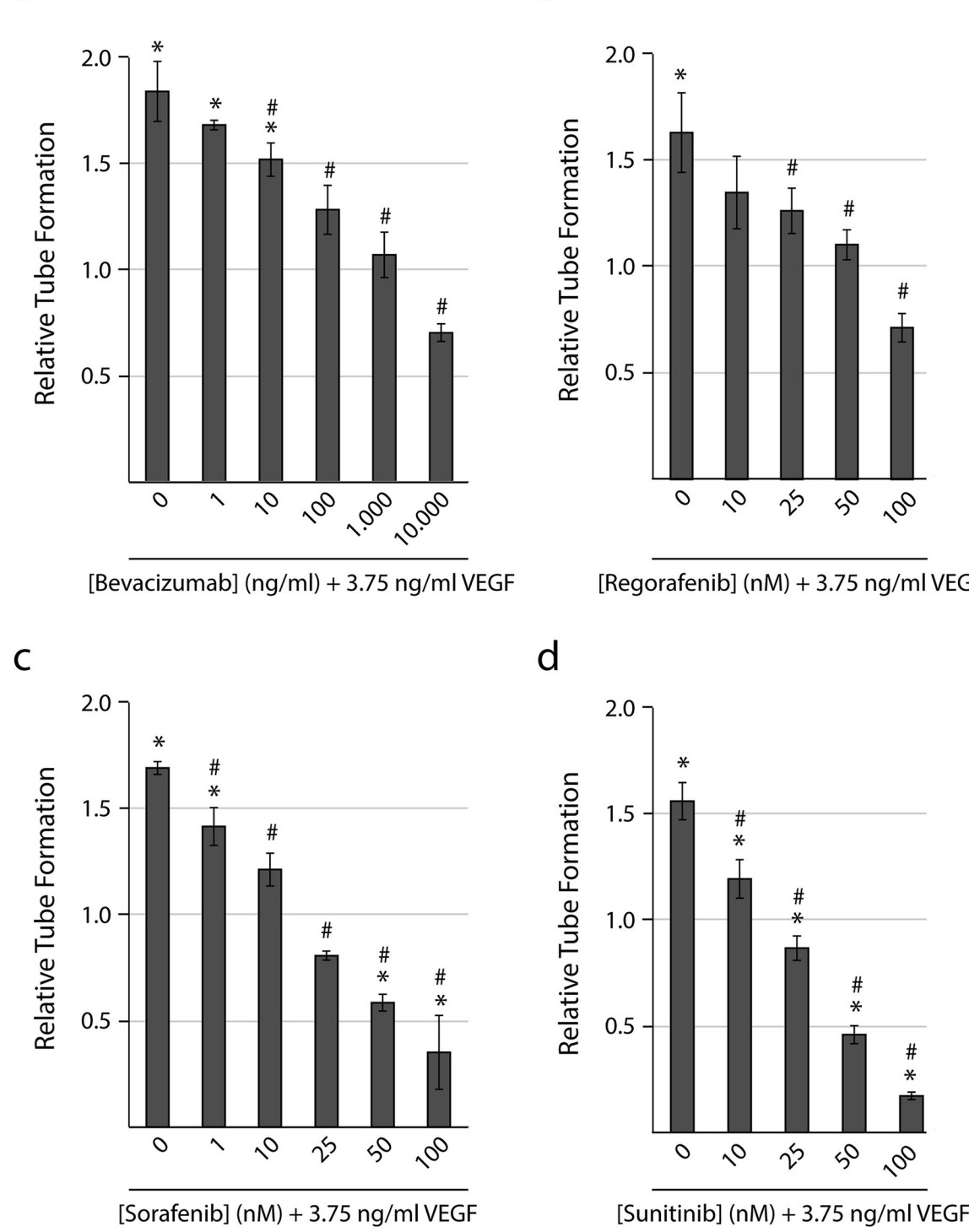

**Fig 3. Dose-response curves following treatment with anti-angiogenic compounds.** HUVECs were seeded on a confluent layer of NHDFs on day 5 and cultured with 3.75 ng/ml VEGF-A$_{165}$ (VEGF) and increasing concentrations of the anti-angiogenic compound from day 7. At the end of the experiment (day 11), the co-cultures were fixed and stained for the endothelial marker PECAM-1 to reveal the tubes. Tube formation is expressed relative to a vehicle treated control. Quantification of tube area in samples treated with increasing concentrations of (a) bevacizumab, (b) regorafenib, (c) sorafenib, and (d) sunitinib. Tubes were quantified with the BioSense Solutions and 2cureX tube algorithm. Two individual experiments were performed for each treatment with representative results displayed. Data

are the average of triplicate samples ± SD. * p<0.05 for treated samples vs. control. # p<0.05 for anti-angiogenic treated vs. VEGF treated samples.

## Increased tube formation did not correlate with secreted levels of antiangiogenic factors

All tumoroids secreted a measurable amount of VEGF-A (Fig 5A). The concentrations ranged from 0.99 ng/ml (patient 9-L) to 3.89 ng/ml (patient 12-L) with a median value of 1.76 ng/ml. The base level of VEGF-A secreted from NHDF, HUVEC cells, and medium was less than 0.1 ng/ml in all investigated control samples. Interestingly, there was no correlation between a high level of secreted VEGF-A and induction of tube formation.

The amount of tumoroid-secreted bFGF and PDGF-BB in the angiogenic tumoroid cultures were also investigated by ELISA (S2 Fig). The bFGF concentrations ranged between 59 and 90 pg/ml, with an average of 69 pg/ml. The PDGF-BB concentrations ranged between 189 pg/ml and 302 pg/ml, with an average of 224 pg/ml. These concentrations were similar to the control samples 80pg/ml and 203pg/ml for bFGF and PDGF-BB respectively (S2 Fig).

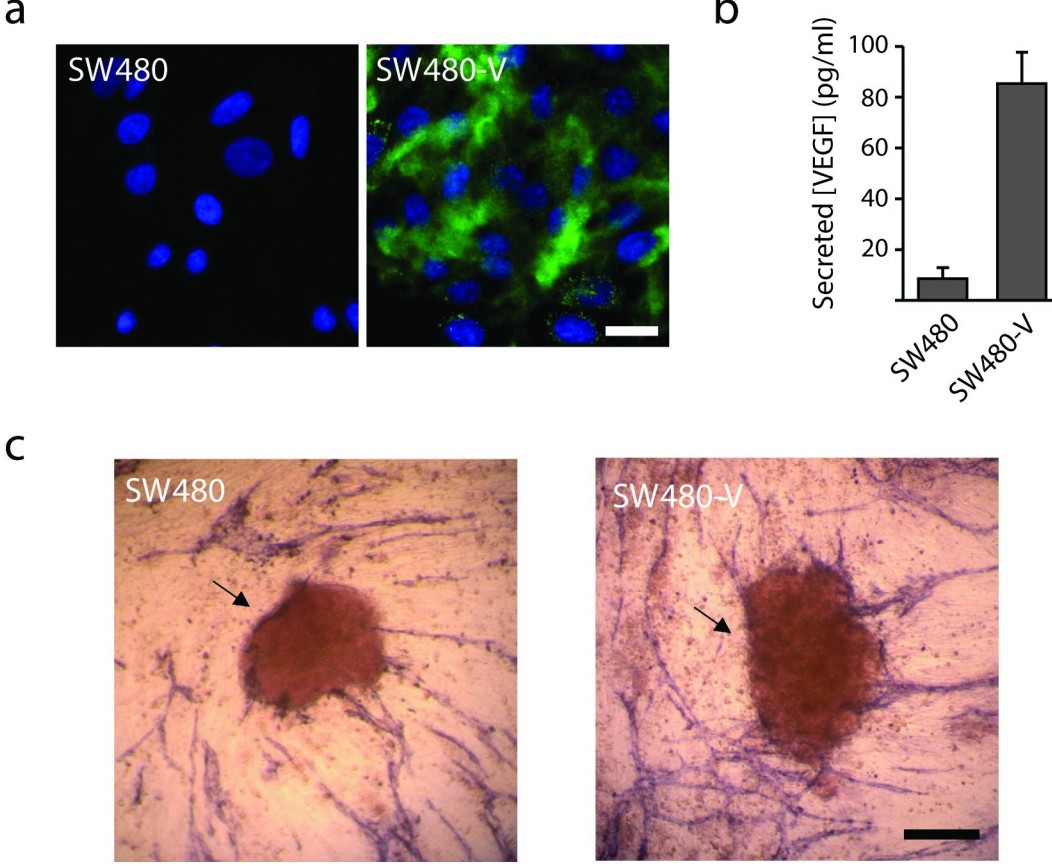

**Fig 4. Vessel growth is directed towards the colon cancer spheroids secreting VEGF- A (VEGF).** (a) SW480 and a SW480 cell line expressing VEGF (SW480-V) were fixed and stained for VEGF (green) and nuclei (DAPI; blue) and imaged by confocal microscopy. Bar 10μm. (b) SW480 and SW480-V cells were grown to confluence and allowed to condition fresh cell culture medium for 48h. The medium was removed and assayed for VEGF by ELISA. Data are the average of three independent experiments ± SEM. (c) Spheroids of SW480 and SW480-V cells were introduced into the co-culture assay of angiogenesis. Vessels were detected by staining for PECAM-1 and imaged by widefield light microscopy. Spheroids (arrows). Bar 250 μm.

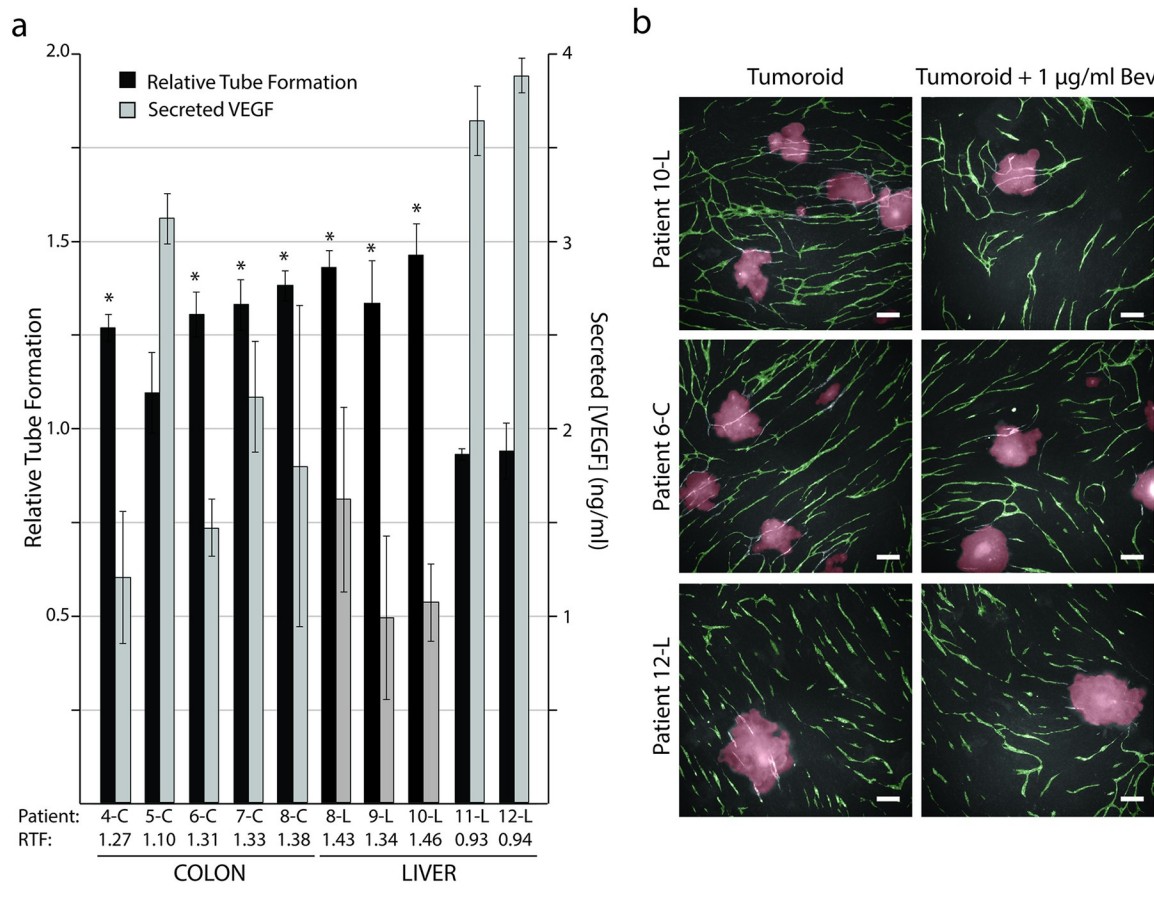

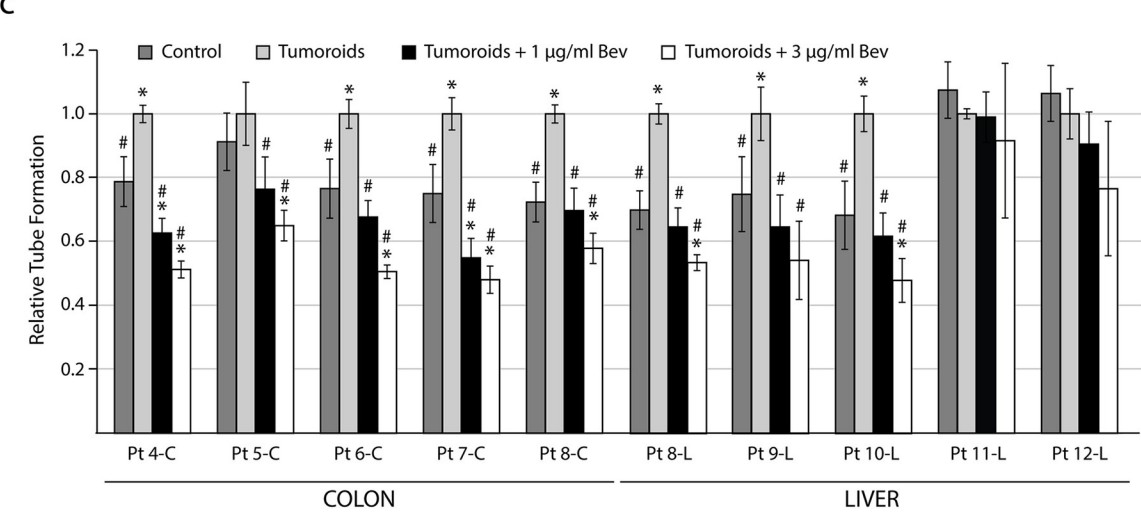

**Fig 5. Patient-derived tumoroids can induce increased tube formation.** Tumoroids were introduced to the CACC assay on day 7. At the end of the experiment (day 11) the co-cultures were fixed and stained for the endothelial marker PECAM-1 to reveal the tubes. Tubes were quantified with the BioSense Solutions and 2cureX tube algorithm. (a) Left axis: Relative tube formation (RTF) for 10 patient-derived tumoroid cultures. Tube formation is expressed relative to control sample without tumoroids. Data are the average of triplicate samples ± SD, * p<0.05 for samples with tumoroids vs. control. Right axis: Tumoroid-secreted VEGF-A (VEGF) measured in culture medium collected from the CACC assay (day 11) and assayed for VEGF by ELISA. Data are the average of triplicate samples ± SD. (b) Representative images (5x objective) of tube formation on day 11 following addition of tumoroids from patient 1x0-L (high induction of tubes), patient 6-C (high to medium induction of tubes) and patient 12-L (no induction of tubes). Images were obtained with the Array Scanner V$^{TI}$ and are shown with green (tubes) and red (tumoroids) overlay from the analysis algorithm. Left

column: Tumoroids in control medium. Right column: Tumoroids treated with 1µg/ml bevacizumab (Bev) from day 7 of the CACC assay. Bar 200 µm. (c) Relative tube formation upon addition of patient-derived tumoroids and treatment with 1 or 3 µg/ml bevacizumab from day 7 to 11 in the CACC assay. Tube formation is expressed relative to samples with tumoroids (no treatment). Pt: Patient * p<0.05 for samples vs. control (no tumoroids, no treatment). # p<0.05 for samples vs. tumoroids (no treatment). Data are the average of triplicate samples ± SD.

## Tumoroid-induced tube formation could be inhibited by bevacizumab

After addition of 1 µg/ml bevacizumab, we found that the tumoroid-induced tube formation in 8 of the 10 cultures was significantly reduced compared with samples containing only tumoroids (p<0.05) (Fig 5B and 5C). In two of the cultures (4-C and 7-C) 1 µg/ml bevacizumab reduced the tube formation to a level significantly lower than the control (no tumoroids, no treatment, p<0.05), and addition of 3 µg/ml bevacizumab resulted in tube formation below the level of the control for 7 out of 10 patients (p<0.05). For patient 11-L and 12-L a significant reduction in tube formation was not observed after addition of bevacizumab.

## Tumoroid-induced tube formation could be inhibited by regorafenib, sorafenib, and sunitinib

Three of the patient-derived tumoroid cultures were chosen to investigate if other anti-angiogenic compounds than bevacizumab could reduce tumoroid-induced tube formation (Fig 6). All of the investigated TKIs induced a significant reduction in tube formation at concentrations of 50 nM and above (p<0.05) when compared with the untreated tumoroid control samples. The most potent inhibitor was sorafenib, which induced a significant reduction in tube formation even at 10 nM (p<0.05). Addition of sorafenib also resulted in tube formation below the level of untreated controls (without tumoroids) at a concentration of 25 nM for culture 6-C (p<0.01) and 10-L (p<0.005). This was observed in all cultures for sunitinib and regorafenib at 50 nM and 100 nM, respectively (p<0.05).

In addition to the ten tumoroid cultures from the 2cureX tumoroid biobank, we also established two new tumoroid cultures from liver metastases biopsies (13-L and 14-L) and investigated the effect of adding these cultures to the CACC assay. On average, 13-L induced a 1.55-fold increase in tube formation compared with the control without tumoroids (p<0.005),

**Table 1. Overview of patients selected for investigations with the cancer angiogenesis co-culture assay.**

| Patient | Sample | Gender | Age (Year) | T | N | M | Histologic subtype |
|---------|--------|--------|-----------|-----|-----|-----|-------------------|
| 4-C | Colon | F | 83 | T2 | N0 | M0 | Glandular |
| 5-C | Colon | M | 76 | T3 | N2 | M0 | Glandular |
| 6-C | Colon | M | 76 | T3 | N1 | M1 | Glandular |
| 7-C | Colon | M | 83 | T3 | N0 | M0 | Glandular |
| 8-C | Colon Biopsy | F | 45 | T4 | N2 | M1 | Mucinous |
| 8-L | Liver Biopsy | F | 45 | - | - | - | N/A |
| 9-L | Liver | F | 51 | - | - | - | Glandular |
| 10-L | Liver | F | 75 | - | - | - | Low Differentiated |
| 11-L | Liver | M | 77 | - | - | - | Glandular |
| 12-L | Liver | M | 59 | - | - | - | Glandular |
| 13-L | Liver Biopsy | N/A | N/A | - | - | - | N/A |
| 14-L | Liver Biopsy | N/A | N/A | - | - | - | N/A |

**Colonic tumors were classified according to the tumour (T), node (N), and metastasis (M) system. C: Colon, N/A: Not available. F: Female, L: Liver, M: Male.**

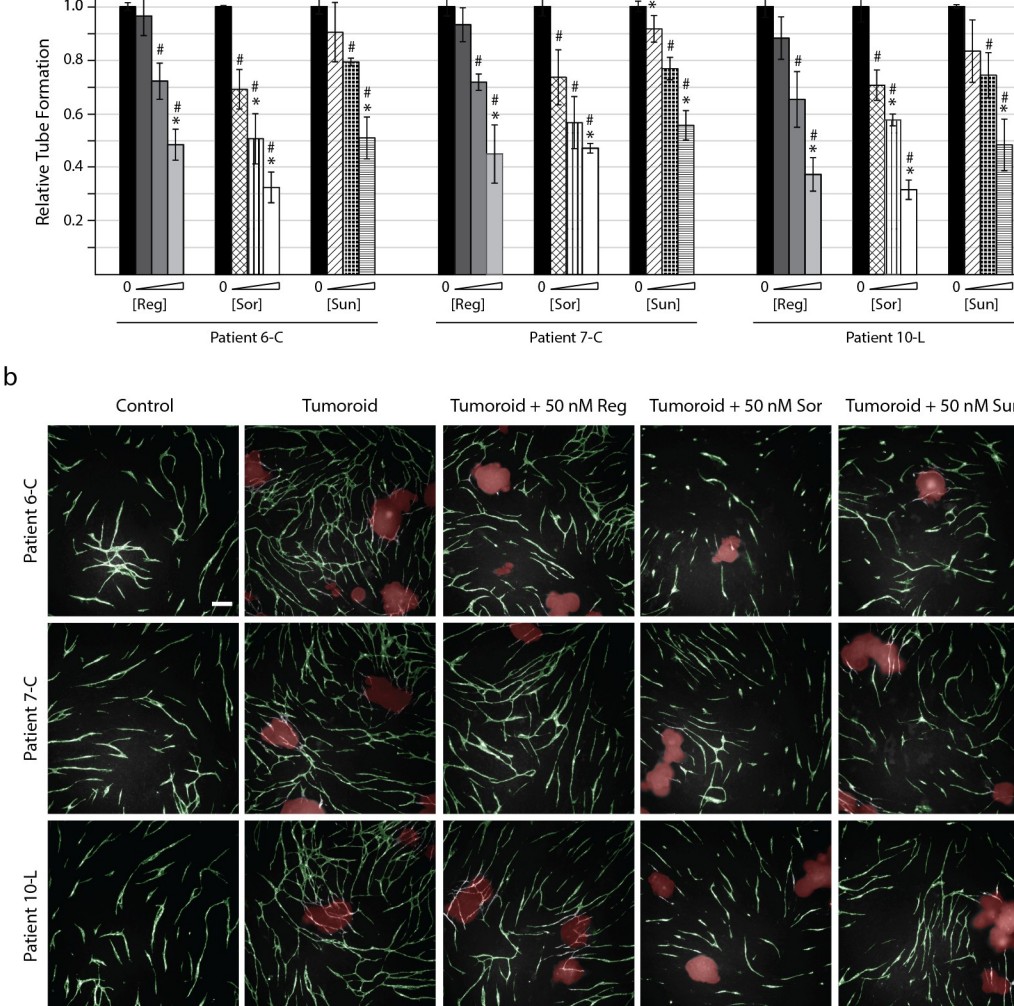

**Fig 6. Anti-angiogenic compounds can abolish tumoroid-induced tube formation.** Tumoroids were introduced to the CACC assay on day 7 ± treatment compounds. At the end of the experiment (day 11) the co-cultures were fixed and stained for the endothelial marker PECAM-1 to reveal tubes. The cultures were treated with increasing concentrations of regorafenib (Reg; 10, 50 and 100 nM), sorafenib (Sor; 10, 25 and 50 nM), and sunitinib (Sun; 10, 25 and 50 nM). (a) Tube formation was quantified with the BioSense Solutions and 2cureX tube algorithm and expressed relative to sample with tumoroids (no treatment). Data are the average of triplicate samples ± SD, * p<0.05 for samples vs. control (no tumoroids, no treatment). # p<0.05 for samples vs. tumoroids (no treatment). (b) Representative images (5x objective) of tube formation on day 11 following addition of tumoroids from patient 6-C, 7-C and 10-L. Images were obtained with the Array Scanner V$^{TI}$ and are shown with green (tubes) and red (tumoroids) overlay from the analysis algorithm. Bar 200 μm.

while 14-L induced a 1.79-fold increase in tube formation (p<0.05) (Fig 7). The amount of tube formation was investigated after addition of increasing concentrations of bevacizumab (0.01, 0.1, and 1 μg/ml), regorafenib (10, 50, and 100 nM) and sorafenib (10, 25 and 50 nM). Interestingly, 13-L tumoroid-induced tube formation was inhibited by less than 20% when treated with 1 μg/ml bevacizumab while tube formation was reduced by 54% in patient 14-L at the same drug concentration.

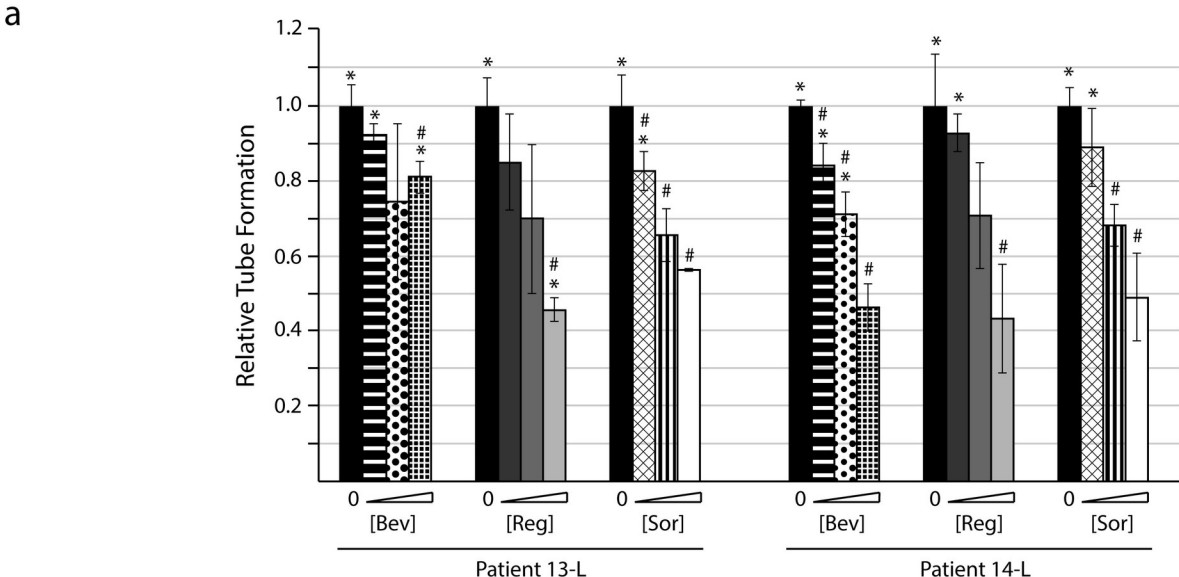

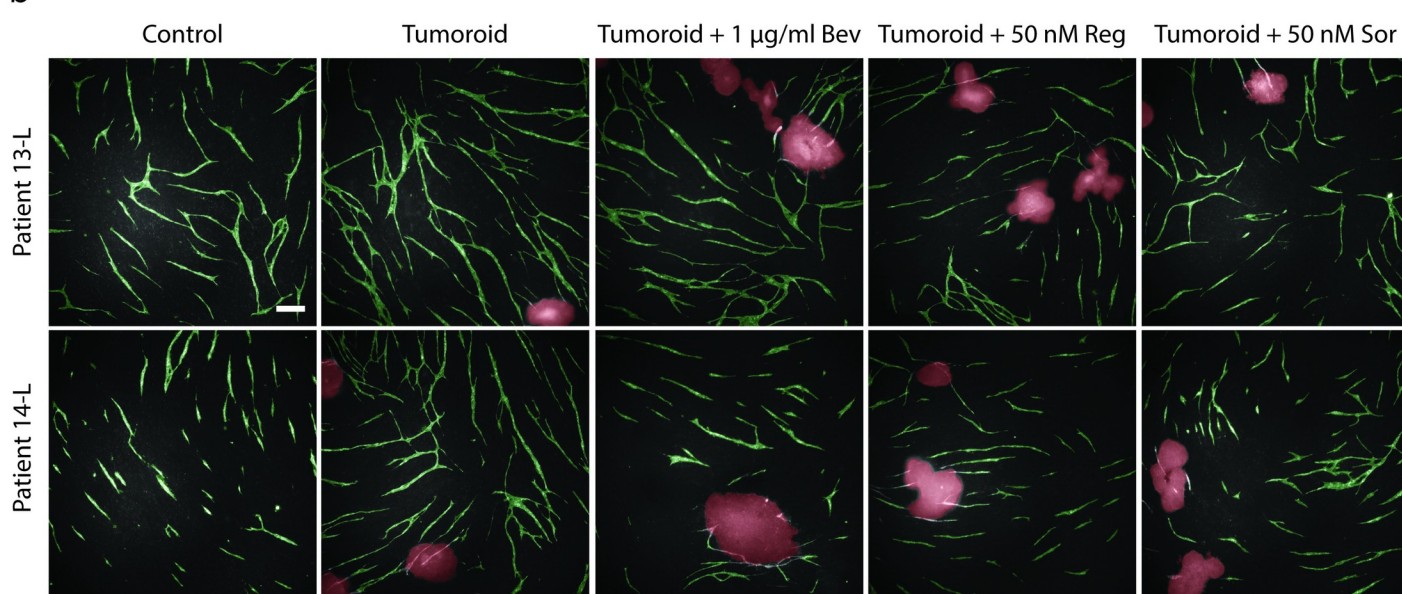

**Fig 7. Tumoroids established from liver metastases express different sensitivities to anti-angiogenic treatments.** Tumoroids were introduced to the CACC assay on day 7 ± treatment compounds. At the end of the experiment (day 11) the co-cultures were fixed and stained for the endothelial marker PECAM-1 to reveal tubes. The cultures were treated with increasing concentrations of bevacizumab (Bev; 0.01, 0.1 and 1 µg/ml), regorafenib (Reg; 10, 50 and 100 nM), and sorafenib (Sor; 10, 25, and 50 nM). (a) Tube formation was quantified with the BioSense Solutions and 2cureX tube algorithm and expressed relative to sample with tumoroids (no treatment). Data are the average of triplicate samples ± SD, * $p < 0.05$ for samples vs. control (no tumoroids, no treatment). # $p < 0.05$ for samples vs. tumoroids (no treatment). (b) Representative images (5x objective) of tube formation obtained with the Array Scanner V$^{TI}$ shown with green (tubes) and red (tumoroids) overlay. Bar 200 µm.

## Inhibition of tube formation was not driven by endothelial cell death

None of the angiogenic inhibitors induced a significant increase in HUVEC or NHDF cell death (active casp-3) at the investigated concentrations (Fig 8). Concurrently, the positive control which was treated with the known cell death inducer staurosporine induced at more than two-fold induction of cell death in both NHDFs and HUVEC covered areas.

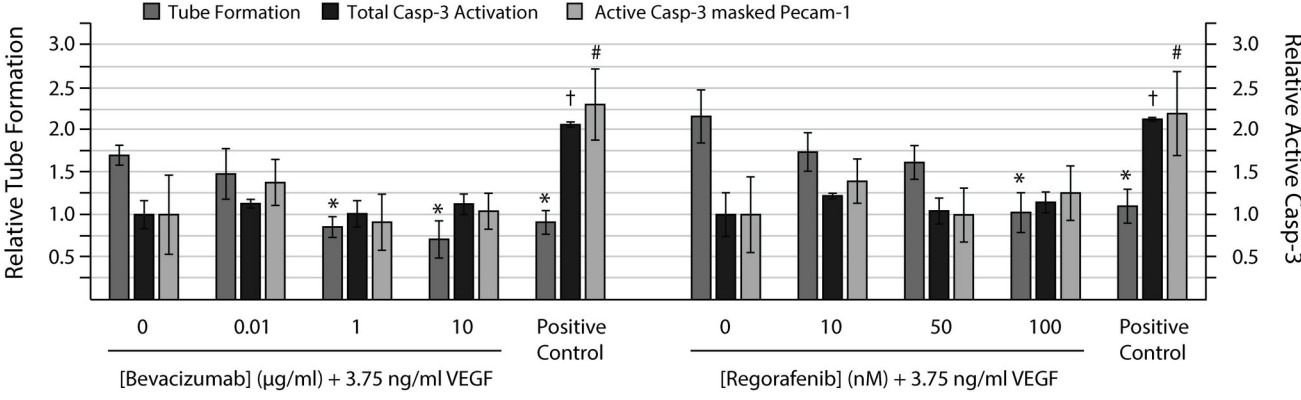

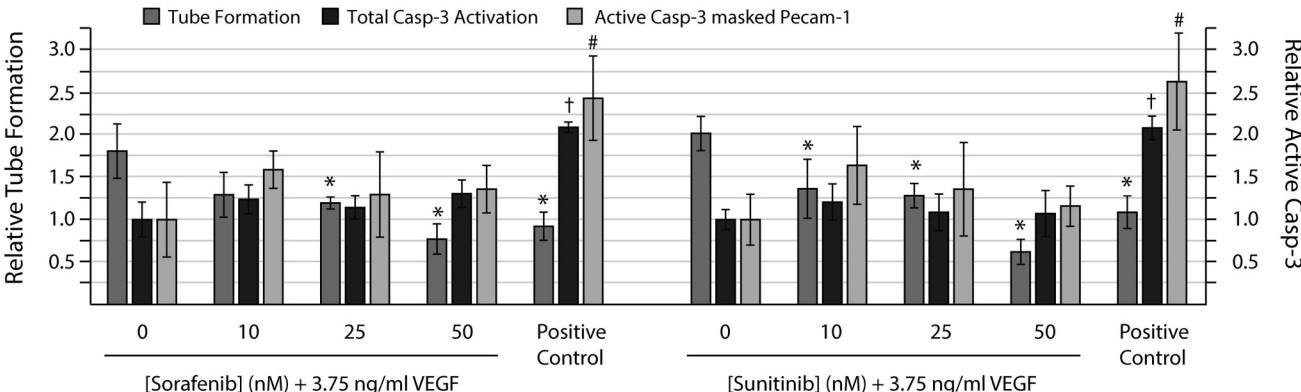

**Fig 8. Anti-angiogenic compounds can abolish tube formation without induction of cell death.** HUVECs were seeded on a confluent layer of NHDFs on day 5 and cultured with 3.75 ng/ml VEGF-A$_{165}$ (VEGF) from day 7 ± increasing concentrations of the anti-angiogenic compounds bevacizumab (0.01, 1 and 10 μg/ml), regorafenib (10, 50 and 100 nM), sorafenib (10, 25 and 50 nM), and sunitinib (10, 25 and 50 nM). At the end of the experiment (day 11), the co-cultures were fixed and stained for the endothelial marker PECAM-1 and the apoptotic marker active caspase-3 (Casp-3) to reveal tubes and apoptotic cells, respectively. Left axis: Relative tube formation. Tube formation is expressed relative to the untreated control. Right axis: Relative active caps-3. Active casp-3 is expressed relative to the VEGF treated sample. Active casp-3 masked PECAM-1: Active casp-3 in areas covered with HUVEC cells. The positive control was treated with 1 μM staurosporine for 6h prior fixation. Tubes were quantified with the 2cureX tube and casp-3 algorithm. Two individual experiments were performed, and representative results are shown. Data are the average of triplicate samples ± SD. *, †, # p<0.05 for anti-angiogenic treated vs. VEGF treated samples with respect to tube formation, total casp-3 activation in the sample and active casp-3 masked PECAM-1, respectively.

## Discussion

In the present paper we determined the angiogenic potential and anti-angiogenic treatment sensitivity of 12 tumoroid cultures established from primary colorectal tumours or liver metastasis, thus opening the door for personalized therapy. They were all evaluated for their sensitivity towards bevacizumab and five of the cultures were also evaluated for their sensitivity towards three different TKIs.

Patient-derived tumoroids and organoids are important 3D model systems since they present individual drug response profiles and can predict the response to chemotherapy with high sensitivity and specificity [39, 44–47]. Such microtumour cultures can be propagated *in vitro* and frozen in biobanks [44, 45, 48]. The main target of most anti-angiogenic agents is the tumour vasculature and accordingly the ECs. These cells have not been incorporated into previous tumoroid and organoid chemosensitivity assays, and prediction of sensitivity to anti-angiogenic agents was therefore not feasible.

Many 3D *in vitro* methods for studying tumour angiogenesis use embedding of cancer cells, cancer cell line spheroids or freshly resected cancer tissue into a matrix gel with or without addition of ECs or stromal cells such as fibroblasts and pericytes [34, 35, 49, 50]. Such assays are adequate models of *in vivo* angiogenesis, but their ability to deliver rapid and quantifiable readouts are inferior since the resulting tube formation and infiltration are scattered throughout the matrix. Our CACC assay is a novel 3D *in vitro* model of tumour angiogenesis employing patient-derived tumoroids co-cultured with tube forming HUVECs and fibroblasts. The confluent layer of fibroblasts is only 3–5 cells in thickness and thereby enables microscopy in a single plane while concurrently allowing the ECs and tumoroids to be cultured in a microenvironment that includes stromal cells and extra cellular matrix (ECM) [38].

The Z' factor is a measure of the quality of a screening assay and ranges from $-\infty$ to 1. An ideal assay would have a Z'-factor of 1, and a screening assay is generally classified as excellent if $1 > Z' \geq 0.5$ [51, 52]. Z' incorporates the dynamic range (the difference between the means of the negative and positive control) and the separation band (the difference between the SD of the negative and positive control). Based on the data from Fig 2, we calculated a Z' factor of 0.54 for the CACC assay meaning that the assay qualifies as an excellent screening assay. An important aspect when developing a screening assay to be utilized in a clinical setting is reproducibility, speed, and cost-effectiveness. The CACC assay was developed using HUVECs as these ECs are widely available and reported to represent a robust source of primary human ECs in the organotypic co-culture assay of angiogenesis [53, 54]. This is supported by the fact that different batches of HUVEC cells demonstrated the same angiogenic potential upon treatment with pro-angiogenic factors. The assay also proved highly reproducible when the same tumoroid cultures were investigated in repeated independent experiments (S3 Fig).

Another important aspect during assay development was to verify that the observed inhibition of tube formation was not mediated by inducing cell death in the ECs. We therefore investigated whether treatment with bevacizumab, regorafenib, sorafenib or sunitinib led to increased apoptosis in both fibroblasts and more importantly the ECs of the CACC assay. The results (Fig 8) show that none of the anti-angiogenic treatments resulted in significant increase in apoptosis in neither the fibroblast nor the ECs at the investigated concentrations. This supports that the treatments reduced tube formation through their anti-angiogenic properties and not by inducing apoptosis in the ECs.

The majority of our results were obtained using biobanked tumoroid cultures. To investigate whether the CACC assay results were affected by the biobanking procedure, we performed the CACC assay on two fresh tumoroid cultures (pt13-L and pt14-L) (Fig 7). The results show that the tube formation with fresh tumoroids was morphologically indistinguishable from the biobanked tumoroid cultures. The two fresh cultures had the highest observed RTF but whether this was due to the age of the tumoroid cultures is unknown. These results emphasize the value of using biobanked tumoroid cultures for assay development. Nine tumoroid cultures were classified as angiogenic and three as non-angiogenic. Tumoroid cultures presented with individual angiogenic profiles since the induced relative tube formation values ranged from 0.93 to 1.79 (Figs 5A and 7). Interestingly we observed that tumoroids deriving from the primary and metastatic lesions from the same patient (pt8) exhibited a similar angiogenic potential. It is rare to obtain samples from both primary and metastatic lesions; however, it would be interesting to investigate the angiogenic potential in more of these dual lesion patients.

The CACC assay only depicts angiogenesis but some tumours use e.g. vessel co-option to supply vessels. Bevacizumab was designed to inhibit sprouting angiogenesis and not tumour co-option of mature peritumoral vessels [31]. In mCRC up to 30% of liver metastases predominantly present with the histopathological growth pattern (HGP) termed replacement-HGP.

This pattern is characterized by co-opting of the peritumoral vessels and may explain why treatment with bevacizumab does not prolong the OS of some patients [31, 55, 56]. A retrospective explorative study investigating the predictive value of HGP showed that the OS was significantly shorter for the cohort of bevacizumab treated patients with replacement-HGP [31]. However, HGP patterns can only be determined on resected liver tumours [56], and is therefore not a suitable biomarker for the choice of anti-angiogenic agents in patients with non-resectable tumours. Possibly, the non-angiogenic tumoroid cultures (patient 5-C, 11-L and 12-L) derive from co-opting tumours and these patients may not benefit from bevacizumab treatment.

The tube formation detected in the CACC assay represents the cumulative effect of both pro- and anti-angiogenic factors in the sample. We observed no correlation between the amount of tumoroid-secreted VEGF-A and induction of tubes (Fig 5A) confirming that VEGF-A is not the only angiogenic factor regulating the process of angiogenesis. These findings are consistent with several clinical studies showing that VEGF-A levels did not correlate with the functional effect of bevacizumab treatment [27, 57–59]. The ELISA assay measured the combined amount of the VEGF-A isoforms and differences between isoform levels within each tumoroid culture are therefore unknown. It would be interesting to investigate how isoform patterns differ between patients and whether this has an impact on the tube induction.

Since the non-angiogenic tumoroids (patient 5-C, 11-L, and 12-L) secreted VEGF-A, the tube formation in these samples are probably inhibited by endogenous anti-angiogenic factors.

The tumoroids from patient 13-L induced the second highest tube formation among the 12 investigated tumoroid cultures. However, the 13-L culture was largely non-sensitive to bevacizumab (Fig 7) suggesting that VEGF-A was probably not a major angiogenic driver for this tumour. Likewise, it is unlikely that the lack of response to bevacizumab was caused by tumour co-option but rather the interplay of several pro-angiogenic factors since the TKIs regorafenib and sorafenib could obliterate tube formation in this culture.

To investigate the potential effect of other angiogenic factors the amount of tumoroid-secreted bFGF and PDGF-BB were also evaluated by ELISA (S2 Fig). The measured concentrations of both bFGF and PDGF-BB for angiogenic cultures were all similar to that of the controls. These results indicate that neither bFGF nor PDGF-BB are likely to have significantly contributed to the higher tube formation observed in the angiogenic cultures. For bFGF this correlates well with published results in a similar tube formation assays where co-cultures of HUVEC and NHDF were treated with increasing concentrations of bFGF. The $EC_{50}$ value of bFGF was found to be 1400 pg/ml [60], which is more than 14 times higher than the highest bFGF concentration found in our angiogenic cultures. We only measured the soluble forms of VEGF-A, bFGF and PDGF-BB and it is unknown whether ECM bound forms are sequestered within the matrix and how they would influence tube formation.

We tested three different TKIs (regorafenib, sorafenib, sunitinib) for their ability to inhibit tube formation. For all investigated tumoroid cultures we saw a dose dependent inhibition of tube formation for each of the TKIs (Figs 6 and 7). The three TKIs tested are multi-kinase inhibitors targeting several of the kinases associated with angiogenesis [22–24]. As discussed above tumoroids from patient 13-L were largely insensitive to bevacizumab but showed sensitivities towards regorafenib and sorafenib which were equal to that of the other investigated tumoroid cultures (Fig 7). This once again emphasises that angiogenesis is regulated by multiple pro-angiogenic factors and suggest that treatment with more broad-spectrum kinase inhibitors in some cases could improve the overall treatment response compared to treatment with more targeted drugs such as the monoclonal antibody bevacizumab.

The present study shows that angiogenesis, as measured in our CACC assay, is a complex interplay between multiple inducers and inhibitors of tube formation and that the assay can

stratify tumoroid cultures as sensitive or resistant to anti-angiogenic therapy thus potentially profile the individual patients´ expected drug responses *in vitro*. The secretion data emphasises that measurements of single drug targets for either monoclonal antibodies or TKIs (e.g. VEGF, bFGF or PDGF-BB) cannot account for the cumulative effect of all present pro- and antiangiogenic factors.

The patients included in this study were all curatively resected, consequently, it was not possible to correlate our results with patient outcome as the patients were not offered anti-angiogenic treatments. Validation of the ability of the test to predict resistance or sensitivity towards a specific anti-angiogenic treatment in the individual patient requires prospective clinical trials comparing the assay results with patient outcome and such trials will hopefully be conducted in the near future.

## Supporting information

**S1 Fig. VEGF secretion from small and large tumoroids.** Tumoroids sized 70–100 μm (n = 40) and 200–400 μm (n = 10) were seeded in a 96-well plate and conditioned media were collected following 7 days of culture to measure the level of tumoroid-secreted VEGF by ELISA. Data are the average of triplicate samples ± SD.
(TIF)

**S2 Fig. Tumoroid-secreted angiogenic factors.** (a) Tumoroid-secreted bFGF measured in culture medium collected from the CACC assay and assayed for bFGF by ELISA. Data are the average of triplicate samples ± SD. (b) Tumoroid-secreted PDGF BB measured in culture medium collected from the CACC assay and assayed for PDGF BB by ELISA. Data are the average of duplicate samples ± SD.
(TIF)

**S3 Fig. The relative tube formation of a tumoroid culture is stable between independent experiments.** Tumoroids were introduced to the CACC assay on day 7. At the end of the experiment (day 11) the co-cultures were fixed and stained for the endothelial marker PECAM-1 to reveal the tubes. Tubes were quantified with the BioSense Solutions and 2cureX tube algorithm. Tube formation is expressed relative to control sample without tumoroids ± SD. Data are the average of 4 independent experiments for Pt 6-C, Pt 7-C and Pt 10-L, and 3 independent experiments for Pt 13-L and 14-L, with triplicate samples in each independent experiment.
(TIF)

## Acknowledgments

The authors thank Tina Lee Brøndum and Sandie Jönch Møller at Bispebjerg Hospital for help with inclusion of patients.

## Author Contributions

**Conceptualization:** Sarah Line Bring Truelsen, Klaus Qvortrup, Ole Thastrup, Henrik Harling, Harry Mellor, Jacob Thastrup.

**Data curation:** Nabi Mousavi, Erik Spillum, Harry Mellor, Jacob Thastrup.

**Formal analysis:** Sarah Line Bring Truelsen, Nabi Mousavi, Haoche Wei, Lucy Harvey, Erik Spillum, Harry Mellor, Jacob Thastrup.

**Funding acquisition:** Sarah Line Bring Truelsen, Erik Spillum, Grith Hagel, Klaus Qvortrup, Ole Thastrup, Henrik Harling, Harry Mellor, Jacob Thastrup.

**Investigation:** Sarah Line Bring Truelsen, Nabi Mousavi, Haoche Wei, Lucy Harvey, Rikke Stausholm.

**Methodology:** Sarah Line Bring Truelsen, Haoche Wei, Lucy Harvey, Erik Spillum, Grith Hagel, Ole Thastrup, Harry Mellor, Jacob Thastrup.

**Project administration:** Ole Thastrup, Harry Mellor.

**Resources:** Ole Thastrup, Henrik Harling, Harry Mellor.

**Software:** Erik Spillum, Jacob Thastrup.

**Supervision:** Klaus Qvortrup, Henrik Harling, Harry Mellor, Jacob Thastrup.

**Validation:** Sarah Line Bring Truelsen, Nabi Mousavi, Haoche Wei, Lucy Harvey, Harry Mellor, Jacob Thastrup.

**Visualization:** Sarah Line Bring Truelsen, Haoche Wei, Lucy Harvey, Harry Mellor, Jacob Thastrup.

**Writing – original draft:** Sarah Line Bring Truelsen, Ole Thastrup, Henrik Harling, Jacob Thastrup.

**Writing – review & editing:** Sarah Line Bring Truelsen, Nabi Mousavi, Haoche Wei, Lucy Harvey, Erik Spillum, Grith Hagel, Klaus Qvortrup, Ole Thastrup, Henrik Harling, Harry Mellor, Jacob Thastrup.

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
