## [Decision Letter · Decision Letter 0]

17 Jul 2020

PONE-D-20-16783

The Cancer Angiogenesis Co-Culture Assay: In vitro quantification of the angiogenic Potential of Tumoroids

PLOS ONE

Dear Dr. Thastrup,

Thank you for submitting your manuscript to PLOS ONE. After careful consideration, we feel that it has merit but does not fully meet PLOS ONE’s publication criteria as it currently stands. Therefore, we invite you to submit a revised version of the manuscript that addresses the points raised during the review process.

Kindly review the attached comments from the reviewers. All reviewers have concerns with the assay itself which must be addressed prior to acceptance.

We look forward to receiving your revised manuscript.

Kind regards,

Christina L Addison, Ph.D.

Academic Editor

PLOS ONE

Journal Requirements:

2.Thank you for including your ethics statement:  "The study protocols were approved by the local ethics committee (protocol no. H-1-2011-125, H-16031106 and S-20170028) and informed consent was obtained from all

patients.

4.Thank you for stating the following in the Financial Disclosure section:

[SLBT, NM, JT and OT was supported by the Innovation Fond Denmark  (http://innovationsfonden.dk/en), grant no. 5184-00101B. SLBT received funding from Agnes & Poul Friis Fund (grant no. 81008-001), C.C. Klestrup & hustrus Henriette Klestrups Mindelegat (grant no. 0660-001), Carl & Ellen Hertz legat til dansk læge- og naturvidenskab (grant no. 7179-2), The Drost Fundation (grant no. 12120-1), and Familien Hede Nielsens Fund (grant no 2017-0423). GH, OT and JT are fulltime employees at 2cureX, Symbion, Denmark (https://www.2curex.com/). 2cureX provided support in the form of salaries for authors GH, OT and JT. The funders had no role in study design, data collection and analysis, decision to publish, or preparation of the manuscript.].   

We note that one or more of the authors are employed by a commercial company: BioSense Solutions

We also note that one or more of the authors is affiliated with the funding organization, indicating the funder may have had some role in the design, data collection, analysis or preparation of your manuscript for publication; in other words, the funder played an indirect role through the participation of the co-authors.

Reviewers' comments:

Reviewer's Responses to Questions

**Comments to the Author**

1. Is the manuscript technically sound, and do the data support the conclusions?

Reviewer #1: No

Reviewer #2: No

Reviewer #3: Partly

2. Has the statistical analysis been performed appropriately and rigorously? 

Reviewer #1: No

Reviewer #2: Yes

Reviewer #3: Yes

3. Have the authors made all data underlying the findings in their manuscript fully available?

Reviewer #1: Yes

Reviewer #2: Yes

Reviewer #3: Yes

4. Is the manuscript presented in an intelligible fashion and written in standard English?

Reviewer #1: Yes

Reviewer #2: Yes

Reviewer #3: Yes

5. Review Comments to the Author

Reviewer #1: The goal of this study is to develop an in vitro assay in which patient-derived tumor cell spheroids are used to assess the ability of EC to form tubes on a fibroblast monolayer. The manuscript is clearly written and provides sufficient easy-to-follow details of the protocol. The protocol itself is a modification of established protocols. The modification here is that tumor spheroids are introduced as a source of angiogenic or anti-angiogenic factors.

As an assay of tubulogenesis this appears to be a good protocol. Notably the Z’ and reproducibility is good. Hence the manuscript describes a potentially valuable assay for use in rapid screening of patient tumor samples and therapeutics, and is thus a valuable contribution to the field.

The manuscript would benefit if some weaknesses (listed below) are addressed:

1) This assay does not measure sprouting angiogenesis as stated multiple times in the manuscript. This is a tubulogenesis assay. Sprouting angiogenesis is new blood vessel sprouting from pre-existing vessels. As stated in the discussion (page 19), Bevacizumab was designed to inhibit sprouting angiogenesis. The manuscript and its title should be modified to make this distinction clear. Furthermore, it is important to address the possibility that the lack of correlation with VEGF levels maybe because this is not an assay of VEGF-dependent sprouting angiogenesis.

2) Importantly the nature of the assay (discussed in point 1 above) is relevant to the extensive discussion and clinical relevance of “HGP” on pages 19 – 20.

3) HUVECs have a very high propensity to form tubes when plated as described. This may account for the small differences seen upon the addition of tumoroids, and perhaps the low sensitivity of this assay.

4) The data presented in most figures are the average of triplicate samples and not the average of multiple experiments. The validity of the results would be greatly enhanced if the experiments were performed more than once. This is particularly relevant since the magnitude of the effects are relatively small.

5) The various patient-derived tumoroids appear to have a range of sensitivity to Bevacizumab in vitro. It would be very valuable to correlate these observations with actual patient-responsiveness information, if accessible.

6) An analysis (or at least a discussion) of the likely targets of the other TKIs tested should be provided.

Reviewer #2: This study is attempting to model the tumor/vasculature microenvironment with tumoroids. The authors examine tube formation of HUVEC induced by patient tumoroids. Surprisingly, they observe no correlation between secreted VEGFA levels and tube formation but still a strong inhibitory effect of Avastin on tube formation across the majority of tumoroids. My main concern is the failure to fully optimize the tumoroid/endothelial assay prior to performing these anti-angiogenic drug screens. Other types of human endothelial cells should be tested before defaulting to umbilical veins cells. Ideally primary endothelial cells would be isolated from resected tumors, but at a minimum, HMEC and other human endothelial lines should be compared. This would greatly strengthen this reviewers confidence in the fidelity of the model to mimic the tumor micro-environment.

Reviewer #3: The manuscript describes the development of "the cancer angiogenesis co-culture assay (CACC)" as an in vitro precision medicine assay. The assay involves quantification of tubule formation and inhibition (in the presence of anti-angiogenesis agents) of tube formation in a co-culture of vascular cells and tumoroids established from primary tumor or metastasis. The manuscript is well written; however details are lacking as mentioned below. The following issues need to be addressed prior to publication.

1. While developing the co-culture assay, information about cell ratio (HUVECs: fibroblasts:tumoroids) missing. Also, what is the composition of the media used for co-culture assay?

2. Did the necrosis core or hypoxia play any role in secretion of VEGF? Did the size of tumor spheroids play any role in tube formation?

3. When developing CACC assay, were the size of tumor spheroids taken into consideration?

4. Fig 2a: Fluorescent images of all conditions need to be provided

5. For investigating the efficacy of angiogenesis inhibitors, why comparison was made with 3.75 ng/ml VEGF treated samples?

6. Why only the concentrations of VEGF and bFGF in the conditioned media was tested? Specially, when no correlation between high VEGF concentration and tubule formation was observed in Fig 5. Concentrations of other pro-angiogenic molecules released by cancer cells need to be evaluated.

7. Most of the anti-angiogenic agents tested are multi-kinase inhibitors. The concentrations of the corresponding growth factors need to be evaluated.

6. PLOS authors have the option to publish the peer review history of their article (what does this mean?). If published, this will include your full peer review and any attached files.

Reviewer #1: No

Reviewer #2: No

Reviewer #3: No

---

## [Author Response · Author response to Decision Letter 0]

16 Jan 2021

Reviewer 1.

1) This assay does not measure sprouting angiogenesis as stated multiple times in the manuscript. This is a tubulogenesis assay. Sprouting angiogenesis is new blood vessel sprouting from pre-existing vessels. As stated in the discussion (page 19), Bevacizumab was designed to inhibit sprouting angiogenesis. The manuscript and its title should be modified to make this distinction clear. Furthermore, it is important to address the possibility that the lack of correlation with VEGF levels maybe because this is not an assay of VEGF-dependent sprouting angiogenesis.

We agree that the Matrigel chord assay is a tubulogenesis assay, where the cells coalesce in a very short amount of time. In the co-culture assay there is some coalescence early on, but then most of the assay is extension of sprouts and branching of existing structures formed early on. And each chord is capped by a tip cell that clearly leads migration and is morphological indistinguishable from the physiological equivalent. Several published studies have shown that the co-culture assay of angiogenesis recapitulates the key aspects of angiogenesis (see reference [36-38,40] in the manuscript). We have updated the introduction and discussion to reflect that this is not a sprouting angiogenesis assay. (p.5 l.93-94, p.20 l.451)

2) Importantly the nature of the assay (discussed in point 1 above) is relevant to the extensive discussion and clinical relevance of “HGP” on pages 19 – 20.

Thank you for pointing out that the HGP and MVD discussion was perhaps a bit to extensive. We have de-emphasised HGP (and removed MVD as it didn’t add much value) in the discussion and focused on how HGP could explain the non-angiogenic tumoroid cultures. (p.20-21 l.456-462)

3) HUVECs have a very high propensity to form tubes when plated as described. This may account for the small differences seen upon the addition of tumoroids, and perhaps the low sensitivity of this assay.

We have included a discussion of why HUVECs were chosen for our assay in the manuscript. (p19 l.425-431)

4) The data presented in most figures are the average of triplicate samples and not the average of multiple experiments. The validity of the results would be greatly enhanced if the experiments were performed more than once. This is particularly relevant since the magnitude of the effects are relatively small.

We have included supplementary figure 3 in which the Relative Tube Formation of the same five tumoroid cultures were tested independently 3-4 times. We believe these experiments demonstrate the reproducibility of the assay. (S3 Fig, p.19 l.430-431

5) The various patient-derived tumoroids appear to have a range of sensitivity to Bevacizumab in vitro. It would be very valuable to correlate these observations with actual patient-responsiveness information, if accessible.

We agree that correlating the CACC results with patient response would be extremely interesting. The patients included in this study were, however, curatively resected and therefore not offered anti-angiogenic treatment. We hope that such a prospective clinical trial can be performed in the near future. We have updated the discussion explaining why a correlation was not performed. (p.23 l.527-528)

6) An analysis (or at least a discussion) of the likely targets of the other TKIs tested should be provided.

We have included more information regarding the tested TKIs and their targets in both the introduction and discussion. Additionally, we have investigated the level of tumoroid secreted PDGF-BB. PDGF-BB is the highest-affinity ligand of PDGFR-β, which is a common target of the three investigated TKIs. (p4 l.74-76, p.15 l.334-338, p.22 l.493-501, p.22 l.503-511, p.23 l. 524-526, S2 Fig.)

Reviewer 2.

This study is attempting to model the tumor/vasculature microenvironment with tumoroids. The authors examine tube formation of HUVEC induced by patient tumoroids. Surprisingly, they observe no correlation between secreted VEGFA levels and tube formation but still a strong inhibitory effect of Avastin on tube formation across the majority of tumoroids. My main concern is the failure to fully optimize the tumoroid/endothelial assay prior to performing these anti-angiogenic drug screens. Other types of human endothelial cells should be tested before defaulting to umbilical veins cells. Ideally primary endothelial cells would be isolated from resected tumors, but at a minimum, HMEC and other human endothelial lines should be compared. This would greatly strengthen this reviewers confidence in the fidelity of the model to mimic the tumor micro-environment.

In the current CACC assay we decided to use an EC cell line to decrease the potential variance that could arise from using ECs extracted from tumour tissue. The aim of this study was to create an assay capable of robustly detecting differences between cancer cells. By using an EC cell line, we feel we have removed a source of variation. Furthermore, HUVECs are widely available and widely used in similar assays. With that being said we agree with the reviewer that testing both primary tumour associated ECs and other EC cell lines would be very interesting, but beyond the scope of this manuscript. We hope to perform a study where the same tumoroids are tested with different sources of ECs both primary and established cell lines and publish these results in a separate paper.

We have included a discussion of why HUVECs were chosen for our assay in the manuscript. (p19 l.425-431)

Reviewer 3.

1. While developing the co-culture assay, information about cell ratio (HUVECs: fibroblasts:tumoroids) missing. Also, what is the composition of the media used for co-culture assay?

We believe we have covered this in the material and methods section (Cancer angiogenesis co-culture assay) already. (p.9-10 l.195-215)

2. Did the necrosis core or hypoxia play any role in secretion of VEGF? Did the size of tumor spheroids play any role in tube formation?

An excellent question. We have not been able to perform the experiments needed to demonstrate hypoxia or apoptosis as a function of tumoroid size. We have included references to papers showing reduced oxygen in the center of larger spheres and included supplementary figure 1 evaluating the VEGF secretion from small 70-100 µum and large 200-400 µm tumoroids. The results show an average 35 fold increase in VEGF secretion between the small and large tumoroids. This to us indicates that the larger tumoroids are indeed experiencing hypoxia. (S1 Fig., p.14 l.295-298)

3. When developing CACC assay, were the size of tumor spheroids taken into consideration?

See question 2 above. (S1 Fig., p.14 l.295-298)

4. Fig 2a: Fluorescent images of all conditions need to be provided

Figure 2 has been updated to show representative images for each VEGF concentration.

5. For investigating the efficacy of angiogenesis inhibitors, why comparison was made with 3.75 ng/ml VEGF treated samples?

The VEGF concentration of 3.75 ng/ml was selected as this is approximately the highest concentration of VEGF we observed secreted by our tumoroids cultures. We have updated the Results section to explain this. (p.13 l.269-270)

6. Why only the concentrations of VEGF and bFGF in the conditioned media was tested? Specially, when no correlation between high VEGF concentration and tubule formation was observed in Fig 5. Concentrations of other pro-angiogenic molecules released by cancer cells need to be evaluated.

We have included a figure with the bFGF secretion results and added ELISA measurements of PDGF-BB levels from the tumoroid cultures (supplementary figure 2) as we believe these three (VEGF, bFGF and PDGF-BB) are key factors in angiogenesis and some of the most relevant ligands targeted by the anti-angiogenic compounds investigated in this manuscript. The list of potentially relevant factors is very long, and it is beyond the scope of this manuscript to investigate them all. (S2 Fig., p.15 l.334-338, p.22 l.493-500)

7. Most of the anti-angiogenic agents tested are multi-kinase inhibitors. The concentrations of the corresponding growth factors need to be evaluated.

See question 6 above. Furthermore, it would be very interesting to try to discern the involved signalling pathways using more specific kinase inhibitors, however, this is unfortunately beyond the scope of this manuscript. (S2 Fig., p.15 l.334-338, p.22 l.493-500)

---

## [Decision Letter · Decision Letter 1]

17 Feb 2021

PONE-D-20-16783R1

The cancer angiogenesis co-culture assay: In vitro quantification of the angiogenic potential of tumoroids

PLOS ONE

Dear Dr. Thastrup

Thank you for submitting your manuscript to PLOS ONE. After careful consideration, we feel that it has merit but does not fully meet PLOS ONE’s publication criteria as it currently stands. Therefore, we invite you to submit a revised version of the manuscript that addresses the points raised during the review process.

Despite some improvements to the manuscript, there are remaining discrepancies in the data that bring the relevancy of the model and its behavior into question. Please note the specific reviewer comments from the most recent round, but which were also raised in previous reviews. Notably issues arising with the lack of correlation of VEGF expression and the tubule formation in the assay as noted by multiple reviewers. It may require you to measure multiple additional angiogenic factors or demonstrate some level of correlation of your assay with angiogenic potential in patient tumors (for example MVD via CD31 or CD34 vessel enumeration in tumor sections by IHC) to satisfy reviewers concerns.

We look forward to receiving your revised manuscript.

Kind regards,

Christina L Addison, Ph.D.

Academic Editor

PLOS ONE

Reviewers' comments:

Reviewer's Responses to Questions

**Comments to the Author**

1. If the authors have adequately addressed your comments raised in a previous round of review and you feel that this manuscript is now acceptable for publication, you may indicate that here to bypass the “Comments to the Author” section, enter your conflict of interest statement in the “Confidential to Editor” section, and submit your "Accept" recommendation.

Reviewer #1: All comments have been addressed

Reviewer #2: (No Response)

2. Is the manuscript technically sound, and do the data support the conclusions?

Reviewer #1: Yes

Reviewer #2: No

3. Has the statistical analysis been performed appropriately and rigorously? 

Reviewer #1: Yes

Reviewer #2: No

4. Have the authors made all data underlying the findings in their manuscript fully available?

Reviewer #1: Yes

Reviewer #2: Yes

5. Is the manuscript presented in an intelligible fashion and written in standard English?

Reviewer #1: Yes

Reviewer #2: Yes

6. Review Comments to the Author

Reviewer #1: The authors have thoughtfully and adequately addressed all of my concerns with edits to the text and a little more supplementary information.

Reviewer #2: The authors conclusion from this paper is “Our data suggests that the CACC assay reflects the angiogenic potential of a patient’s tumour and can quantify its sensitivity towards different anti-angiogenic agents.”

I still don’t feel their data supports that conclusion. VEGF is the best characterized and most established angiogenic factor. In this report, tumoroids that secrete high levels of VEGF don’t show higher levels of tube formation. Additionally, tumoroids that secrete high levels of VEGF don’t have suppressed tube formation in response to anti-VEGF antibodies. However, tumors that don’t secrete VEGF do have suppressed tube formation in response to anti-VEGF anti-bodies. How can anyone conclude that this assay could reflect the angiogenic potential of the patient’s tumor is puzzling. The authors don’t know the levels of angiogenic factors in the patients nor their responsiveness to anti-angiogenic therapy. So there is no ability to correlate anything to the patient. Likewise, there is no ability to correlate in vitro activities since none of the investigated angiogenic factors secreted by the tumoroids correlated well with HUVEC tube formation nor their responsiveness to the specific inhibitors for that specific factor. One can hand wave about the angiogenic balance of stimulators and inhibitors, but without a demonstration of specific correlations, one is left to consider the possibility that the assay is flawed due to the endothelial cell type used or the other elements of the methods.

7. PLOS authors have the option to publish the peer review history of their article (what does this mean?). If published, this will include your full peer review and any attached files.

Reviewer #1: No

Reviewer #2: No

---

## [Author Response · Author response to Decision Letter 1]

15 May 2021

Dear Christina,

Thank you for the feedback to our manuscript. We have thoroughly reviewed our manuscript following the comments from reviewer 2. In our opinion the most “problematic” conclusion was in the abstract and final part of the discussion (line 525). We agree that the last sentence in the original abstract was a bit optimistic. Consequently, we have revised the manuscript and emphasized that a clinical verification is needed to prove the predictive value of our assay. We also point to the last portion of the Discussion (lines 529-533) where we also write that validation of the assay will require a clinical trial. 

The field of angiogenesis is currently challenged by the fact that no combination of angiogenic factors and/or pathological measurements is able to predict patient response to anti-angiogenic treatment such as Bevacizumab. Consequently, verifying the validity of our assay against known angiogenic factors could prove an impossible task. The observation that our results do not correlate with individual angiogenic factors (such as VEGF) (just like in the clinic) leads us to speculate that our assay could potentially recapture some of the biological complexity of tumour angiogenesis. As written above and in our manuscript (and now emphasized in the updated abstract) the only way to verify this correlation is by conducting a clinical trial which is beyond the scope of this manuscript.

We hope that the changes and the above explanation is sufficient to satisfy you and reviewer 2.

Best regards,

Jacob Thastrup

---

## [Editor Report · Decision Letter 2]

2 Jun 2021

The cancer angiogenesis co-culture assay: In vitro quantification of the angiogenic potential of tumoroids

PONE-D-20-16783R2

Dear Dr. Thastrup

We’re pleased to inform you that your manuscript has been judged scientifically suitable for publication and will be formally accepted for publication once it meets all outstanding technical requirements.

Kind regards,

Christina L Addison, Ph.D.

Academic Editor

PLOS ONE
---

## [Editor Report · Acceptance letter]

28 Jun 2021

PONE-D-20-16783R2 

The cancer angiogenesis co-culture assay: *In vitro* quantification of the angiogenic potential of tumoroids 

Dear Dr. Thastrup:

I'm pleased to inform you that your manuscript has been deemed suitable for publication in PLOS ONE. Congratulations! Your manuscript is now with our production department. 

Kind regards, 

on behalf of

Dr. Christina L Addison 

Academic Editor

PLOS ONE